# Leadership in Construction: A Scientometric Review

**Wang Peng** [1,2,*], **Nuzul Azam Haron** [1], **Aidi Hizami Alias** [1] **and Teik Hua Law** [1]

[1] Department of Civil Engineering, Faculty of Engineering, University Putra Malaysia (UPM), Serdang 43400, Malaysia

[2] Airport College, Binzhou University, Binzhou 256603, China

[*] Correspondence: gs58114@student.upm.edu.my

**Abstract:** Leadership plays an increasingly important role in construction projects, and numerous research studies have been conducted. This study aims to identify the structure evolution development trends of this knowledge domain using visualisation analysis with CiteSpace. A total of 1789 peer-reviewed articles are collected from Scopus and the WoS core collection database to conduct a scientometric analysis. The results indicate that the US dominates this field and that institutions from Australia have made greater contributions. However, international cooperation in this area is not active. A total of eight co-citation clusters were identified, and the research of leadership in construction primarily focused on the topics of transactional leadership, safety leadership, team performance, leadership interaction processes and actual leader behaviour. The keywords co-occurrence evolution analysis was also conducted to provide a holistic knowledge map. Based on the development of this field and its current status, we propose trends and innovative research areas for future research. The findings in this research would help scholars to understand the structure and future trends of this field. Meanwhile, the research results would provide a reference for construction enterprises to formulate project manager competency criteria.

**Keywords:** construction industry; leadership; research trend; scientometrics; visualisation

## 1. Introduction

Research into leadership has a long history. As an extensively explored theory, leadership has gained the attention of scholars worldwide, resulting in a wide variety of qualitative and quantitative approaches. Some scholars conceptualise leadership as a behaviour or personality, while others view it from a social information processing standpoint [1]. For many years, the importance of leadership was ignored in construction project management, as traditional researchers in this field mainly focused on the technology of the project [2,3].

The research paradigm of leadership in the construction industry began to change at the end of the 20th century. Across a broad research scope, early researchers conducted some interesting cross-sectional studies. For example, Sui Pheng and Lee [4] attempted to construct a managerial grid framework with an ancient Chinese strategy to develop the leadership of construction project management, while Weingardt [5] encouraged engineers to become more involved in leadership on all levels and to broaden their impacts on public policy. During this period, scholars realised that construction projects are people-oriented and that effective leadership is inevitably related to project success [6,7].

Since the beginning of the 21st century, leadership has been documented as an important skill of effective project leaders [8] and engineering management students [9] to accompany their technical skills. The behaviours of the project leader were found to be more significant in the prediction of project performance than the other team members' characteristics [10]. Some quantitative and qualitative methods are gradually being introduced in the literature on leadership in construction projects. Liu et al. [11] developed a model to examine the power structures of the leaders of construction projects, while

Fellows et al. [12] determined the leadership style and power relations. Odusami et al. [8] analysed the data from 60 project team leaders and concluded that project performance is significantly affected by the project manager's leadership style. With a rapidly growing body of leadership research in construction projects, a number of studies have been conducted to determine the factors that contribute to an ideal construction project manager's leadership style from a variety of perspectives [13–17]. Since the work of Turner and Müller [17], the study of the leadership of project managers has gained new momentum. Turner and Müller suggested that research should pay particular attention to the project manager's competence and leadership style in relation to project success. For in-depth research, Müller and Turner [18] conducted qualitative and quantitative studies and concluded that the project manager's leadership style influences project success and that different leadership styles are appropriate for different types of projects.

Over the past two decades, leadership has been regarded as a significant cause of project success or failure [19], and the personal and interpersonal factors of project managers that contribute to project performance have been investigated to some extent [20–24]. As shown in Figure 1, since the first literature appeared in 1992, the number of studies has increased every year. Against this background, several studies have reviewed and analysed the existing literature. Toor and Ofori [22] selected 49 studies and summarised the empirical research on leadership in the construction industry. Simmons et al. [23] conducted a critical review that predicted some future trends. In Graham's [24] systematic review study, a computerised technique was used for bibliometric analysis to develop clear criteria to inform a thematic analysis and compare the analyses between multiple authors. Some other studies remain narrow in focus, dealing only with a certain branch of leadership theory in the construction industry [25–27]. Most studies, however, rely on the subjective judgment of experts, which leads to a lack of quantitative bibliometric analysis in this field. Although there are increasing numbers of publications in this field, little is known about the overall structure of the knowledge landscape.

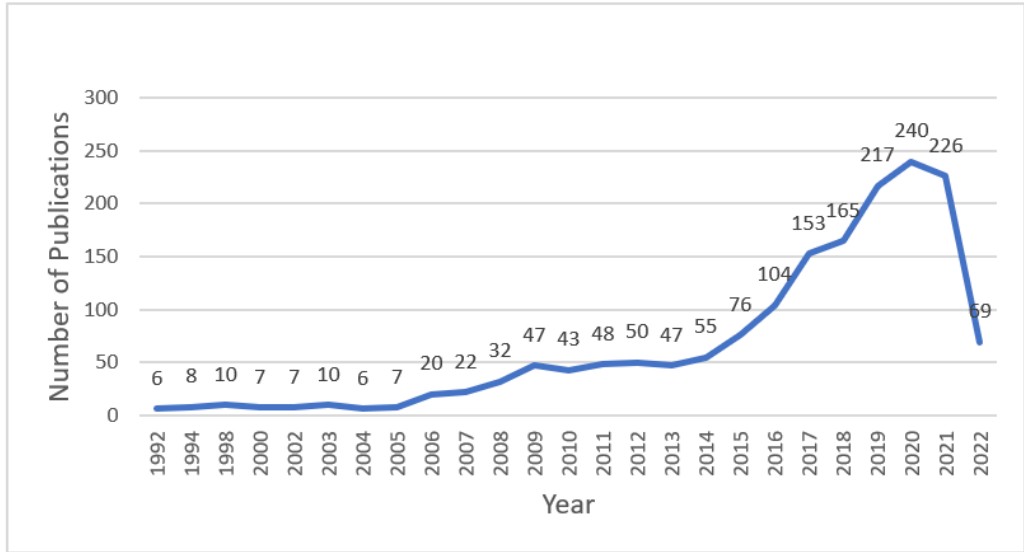

**Figure 1.** The number of published papers on the topic "leadership in the construction industry" or "leadership in construction projects" (1992–2022). Source: Web of Science.

Scientometrics is a branch of informatics that was defined by Nalimov and Mul'chenko [28] as "the quantitative methods of the research on the development of science as an informational process". It can be considered as the quantitative analysis of patterns in the scientific literature to map knowledge structures and predict emerging trends in a research field. Compared with traditional bibliometrics, scientometrics provides in-depth quantitative analysis of the bibliographic information, such as countries, institutions, authors, keywords, and references [29]. The applications of the scientometrics software reduce the impact of a

researcher's subjective opinions on document records' retrieval and screening. By using visualisation tools such as CiteSpace [30], VOSviewer [31] and Histcite [32], the knowledge structure of a scientific research field can be clearly presented. The scientometrics approach is the best way to explore research trends and key areas of study over time [33,34]. Therefore, various visualisation tools have been widely used in the field of technology management analysis [35]. By comparing the features of these software, we found that CiteSpace, which can visualise the networks of cooperation and the development of a research area over time, had the most features and was the most used.

Despite the popularity of visualising the literature, we searched Scopus, WoS and Google Scholar and found that there are still no papers on the current state of leadership research in construction. With the extensive investigation of a field of study, it is necessary to conduct a systematic analysis that provides current conditions and potential future trends. To fill this gap, this study, which attempts to conduct a scientometric review of the scientific literature relating to leadership in construction, is guided by the following key goals: first, to determine the main contributing forces in this research area at the level of countries, institutions and contributing authors; second, to illustrate the intellectual structure and its evolution in the field; third, to identify the disciplines and topics involved in the field; and fourth, to understand emerging trends in the future.

## 2. Methodology

To make the bibliographic data more comprehensive, we selected two world-leading research databases, Scopus and the Web of Science Core Collection, as this article's data source. Scopus is currently the largest database of peer-reviewed literature, containing over 23,452 peer-reviewed journals from 7000 publishers. In comparison, the Web of Science (WoS) core collection database covers papers from approximately 12,000 leading journals. These two databases have been the traditional sources for most major scientometrics studies [36]. To make the data more concentrated, we used "leadership" in the title as the search condition, limiting the search results to construction projects. Therefore, the approach for bibliographic retrieval is as follows: 'TITLE (leadership) AND ALL (construction project)' in Scopus and "(TI = (leadership)) AND ALL = (construction project)" in WoS. There is no limit to the timespan of the retrieved literature. The retrieval time is 27 August 2022. A total of 2350 document results were found in Scopus, and 228 were found in WoS. Peer-reviewed articles are considered to be more representative [37]. Therefore, the bibliographic search was limited to articles and reviews, and non-English publications were excluded. This process resulted in the removal of 657 records, leaving 1737 records from Scopus and 184 from WoS. The two databases are overlapping. After removing duplicates, a final sample of 1789 records was identified.

The scientometric software CiteSpace was used to analyse the data and generate visualised results. The main purpose of this software is to analyse and illustrate a knowledge domain, which is a broadly defined concept that covers a scientific field, a research area or a scientific discipline and is usually represented by a set of bibliographic records of relevant publications. CiteSpace is a widely used scientometric software that makes it easier for users to identify pivotal points [38]. The pivotal points constitute the basic framework of CiteSpace visualisation graphs, including authors, institutions, countries, terms, keywords, categories, cited authors and cited bibliography. The foundation of CiteSpace is network analysis and visualisation. The intellectual landscape of a knowledge domain, the questions that academics have been focused on and the methods and tools they use to achieve their goals are easy to explore through network analysis models and visualisation.

As shown in Figure 2, after the inclusion of the data, there were three types of bibliometric techniques applied in this study: collaboration analysis, co-citation analysis and keyword co-occurrence analysis. In a knowledge domain, it is essential to understand the process of knowledge diffusion. In the selected dataset, each bibliographic record contains the information of the authors and their addresses. CiteSpace analysed each record and identified the collaboration countries, institutes and authors in the same article. The coun-

tries, institutes and authors connected to generate the collaboration network. As shown in this study, the collaboration network can provide a helpful guide for potential cooperation in this research field. Co-citation was first proposed as a new measure of two articles in the early 1970s [39] and has been long considered an effective tool for mapping the intellectual structure of science. Furthermore, cluster analysis was performed based on the co-citation analysis results in this study. The research hotspots in this field in various periods were identified. Co-word analysis was developed in the 1980s. CiteSpace acquires a keyword co-occurrence matrix in the data and detects the citation bursts in the knowledge domain, which is an effective way to trace the research trend and predict future research.

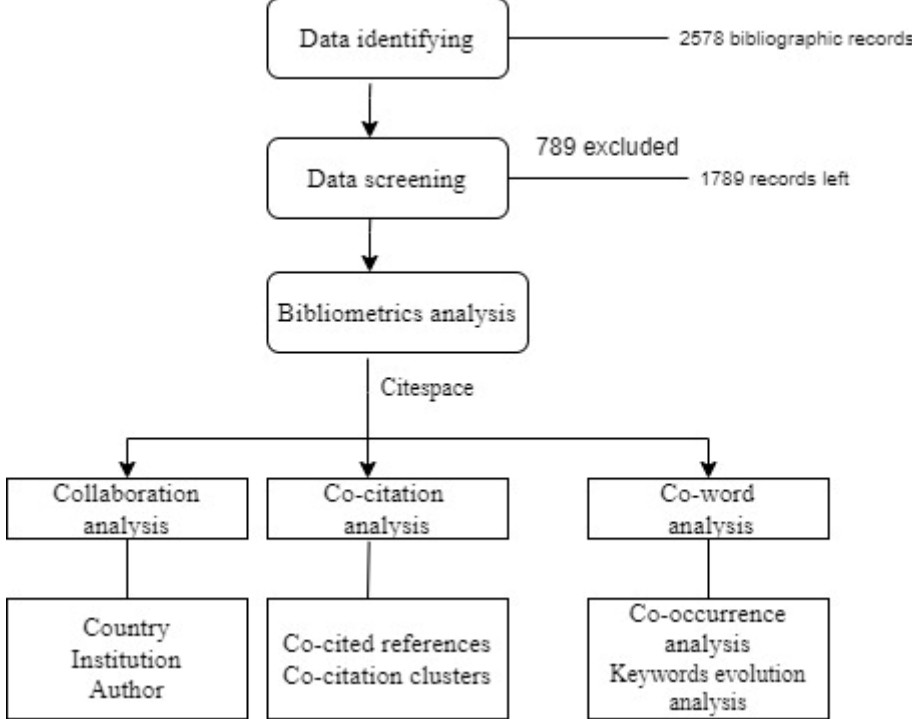

**Figure 2.** Research framework.

### 3. Results

*3.1. Collaboration Analysis*

3.1.1. Country/Region Collaboration Network

As shown in Figures 3 and 4, a country/region collaboration network, which showed the spatial distribution of publications quantitatively, was created based on the number of publications and betweenness centralities. The betweenness centrality refers to the number of times that a node acts as the shortest path between two other nodes. The higher the value, the higher the contribution of a node to make connections with other nodes in a network. The network of collaborating countries contained 118 nodes and 164 links between 1987 to 2022. The country with the highest number of publications is the US, with 418 papers, and the US also contributed the second highest centrality number (0.21), making it the most active country in terms of research in construction leadership, with great contributions to international cooperation (Table 1). The publication records of the UK, China and Australia were 240, 210 and 189, respectively. However, the centrality of China was lower than that of the two other countries. Moreover, the United States and the United Kingdom were the first two countries to start research in this field, nearly ten years earlier than other countries. Research in countries such as Australia, Canada and New Zealand began in the late 1990s. With the continuous development of this research field, scholars from new countries contribute every year. Overall, the United States is leading in this field of research, and the US also has the highest level of burst strength (27.84). Although China has about the same number of publications as Australia and the UK, it is still less influential

than these two countries. With the development of the construction industry, increasing numbers of scholars in developing countries have focused on this field.

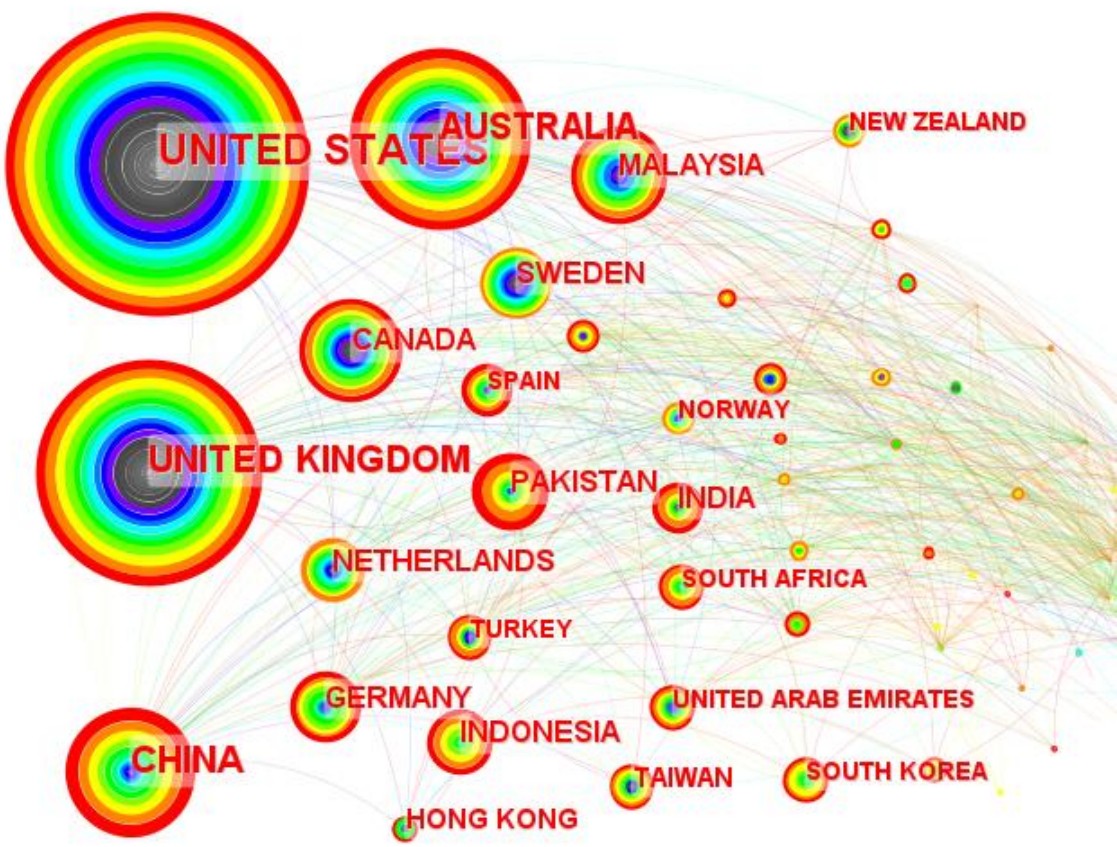

**Figure 3.** Country/region collaboration network.

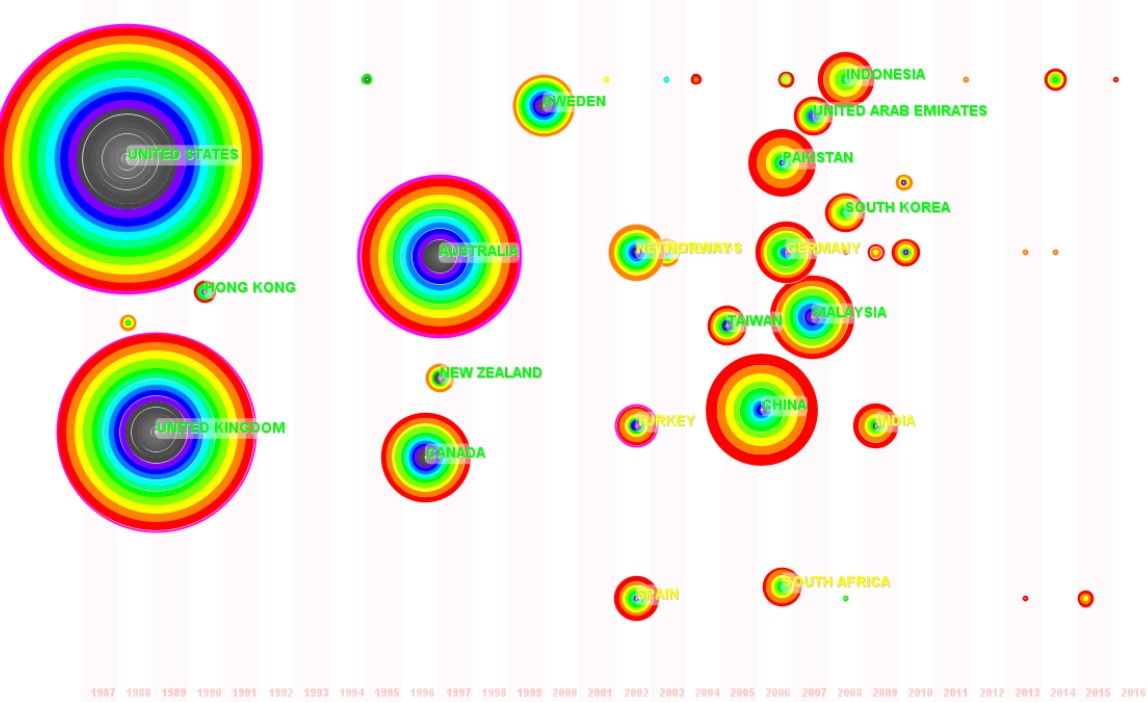

**Figure 4.** Country/region collaboration network: a timezone view.

**Table 1.** Top 10 countries/regions based on the number of publications.

| Country | Publications | Centrality | Country | Publications | Centrality |
|---|---|---|---|---|---|
| US | 418 | 0.21 | Pakistan | 82 | 0.03 |
| UK | 240 | 0.18 | Canada | 73 | 0.07 |
| China | 210 | 0.09 | Germany | 59 | 0.04 |
| Australia | 189 | 0.25 | Indonesia | 57 | 0.01 |
| Malaysia | 90 | 0.07 | Netherland | 51 | 0.04 |

3.1.2. Institution Collaboration Network

As shown in Figure 5, the network of collaborating institutions contained 1430 nodes and 1731 links, with a lower density of 0.0017, and the links between the nodes indicate the cooperation between the involved institutions. The relatively loose structure indicated a low level of institutional cooperation in this field of research. Table 2 lists the top 20 institutions that contributed the most publications, such as RMIT University (25 articles), University of Johannesburg (24 articles), Tongji University (21 articles), Curtin University (20 articles), Tsinghua University (20 articles) and The Hong Kong Polytechnic University (18 articles). It is clear that Australian universities are the largest contributors to research on leadership in construction, with six institutions (RMIT University, Curtin University, Queensland University of Technology, Deakin University and The University of Auckland, ranking 1, 4, 7, 16 and 20, respectively). Although a research group is gradually forming, there are still some institutions without external collaboration. Among all the institutions, only RMIT University and Curtin University have a betweenness centrality exceeding 0.05. This indicates a lack of close cooperation between institutions. The red in Figure 5 indicates the citation burst strength. The top-ranked item by bursts is Curtin University, with bursts of 3.57. The second one is RMIT University, with bursts of 3.24. RMIT University has a high number of publications, a high burst strength and a high betweenness centrality, which indicates its significant contribution to this research field.

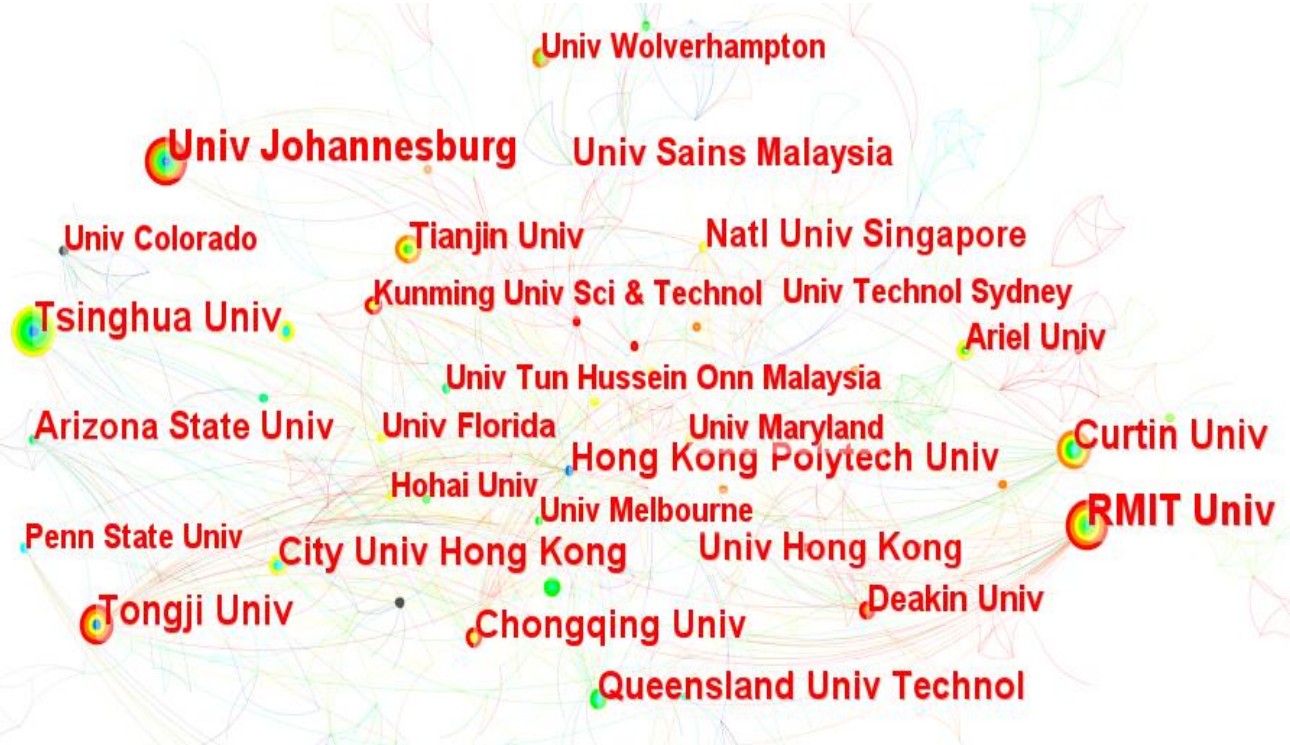

**Figure 5.** Institution collaboration network.

**Table 2.** Top 20 institutions based on the number of publications.

| Institution | Publications | Centrality | Country/Region |
| --- | --- | --- | --- |
| RMIT University | 25 | 0.06 | Australia |
| University of Johannesburg | 24 | 0.01 | South Africa |
| Tongji University | 21 | 0.05 | China |
| Curtin University | 20 | 0.08 | Australia |
| Tsinghua University | 20 | 0.03 | China |
| The Hong Kong Polytechnic University | 18 | 0.04 | Hong Kong |
| Queensland University of Technology | 18 | 0.04 | Australia |
| City University of Hong Kong | 17 | 0.05 | Hong Kong |
| Chongqing University | 17 | 0.04 | China |
| National University of Singapore | 15 | 0.05 | Singapore |
| University of Science Malaysia | 15 | 0.02 | Malaysia |
| The University of Hong Kong | 15 | 0.02 | Hong Kong |
| Arizona State University | 10 | 0.02 | US |
| Tianjin University | 14 | 0.03 | China |
| Ariel University | 13 | 0.00 | Israel |
| Deakin University | 13 | 0.04 | Australia |
| University of Florida | 12 | 0.04 | US |
| University of Colorado | 8 | 0.02 | US |
| University of Wolverhampton | 11 | 0.00 | UK |
| The University of Auckland | 9 | 0.01 | Australia |

### 3.1.3. Author Collaboration Network

The contributions of the authors were also identified. As shown in Figure 6, a loose-structure network of collaborating authors consisted of 4124 nodes and 7884 links. There is a large number of scholars and extensive collaborations in this field. As shown in Table 3, the most productive authors were Shamas-ur-Rehman Toor and Lianying Zhang, who both published 11 articles in this field. However, most authors received a centrality number of 0. The betweenness centrality of a node in the network measures the importance of a node's role as a bridge in the network. Since the system only retains two decimal places, the number 0 means that most of the authors, as nodes, had a centrality of less than 0.01, which indicates that the cooperation in this area is dominated by small teams. Further analysis revealed some communication between the research teams, but there is still a lack of key connecting hubs between different research groups.

**Table 3.** Top 10 authors based on the number of publications.

| Author | Publications | Centrality | Author | Publications | Centrality |
| --- | --- | --- | --- | --- | --- |
| Shamas-ur-Rehman Toor | 11 | 0 | Ralf Müller | 9 | 0.01 |
| Lianying Zhang | 11 | 0.02 | Sadaf Iqbal | 9 | 0.02 |
| Ying Chen | 10 | 0 | Junwei Zheng | 8 | 0 |
| Umer Zaman | 10 | 0 | George Ofori | 8 | 0 |
| Yanfei Wang | 9 | 0.02 | Li H | 8 | 0.02 |

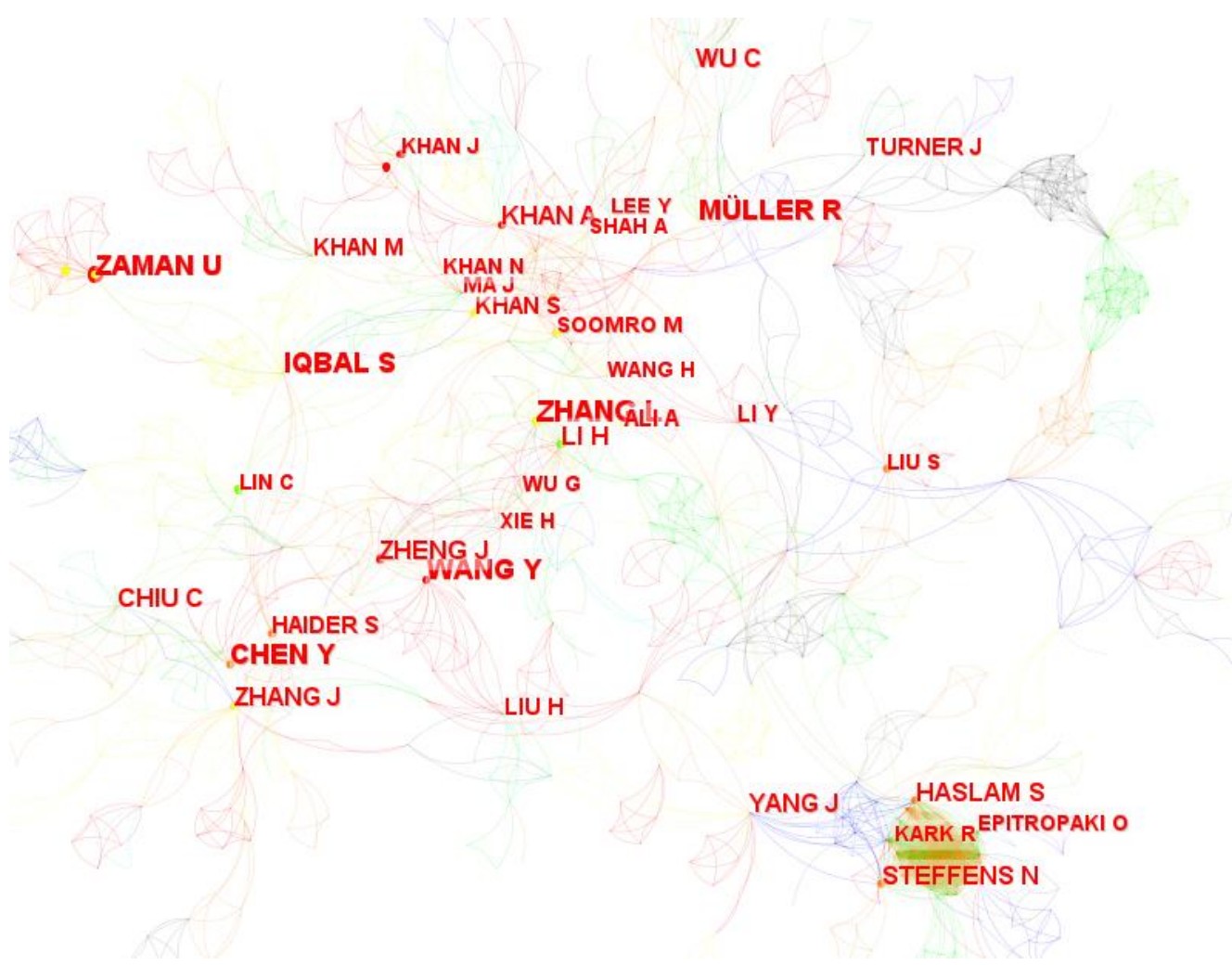

**Figure 6.** Author collaboration network.

*3.2. Co-Citation Analysis*

3.2.1. Author Co-Citation Analysis

As illustrated in Figure 7, the author co-citation network with 1147 nodes and 7737 links was conducted to clarify the essential information of authors and their relationships. There are a few significant nodes in the network, which shows that the leadership research basis in the construction industry is relatively concentrated. The top 10 authors in terms of co-citation frequency, burst strength and centrality are listed in Table 4. Bernard M. Bass has the highest frequency of co-citation and the second-highest centrality. Therefore, it can be said that his research is the foundation of this field. Bernard M. Bass and Bruce Avolio have worked closely together and have contributed significantly to transformational leadership research [40–43]. Both of them are also the authors with the highest centrality values, which shows that transformational leadership has received more attention in construction leadership research. The third most co-cited author is Philip M Podsakoff, whose research interest is the intersection of organisational behaviour and leadership, along with Robert J. House. Among the ten most co-cited authors, Joseph F. Hair and Claes Fornell are interested in multivariate analysis and structural equation models. Moreover, the co-citation of Jörg Henseler, who also focuses on research methods, burst in the year 2019, which indicates the growing use of quantitative methods in construction leadership research.

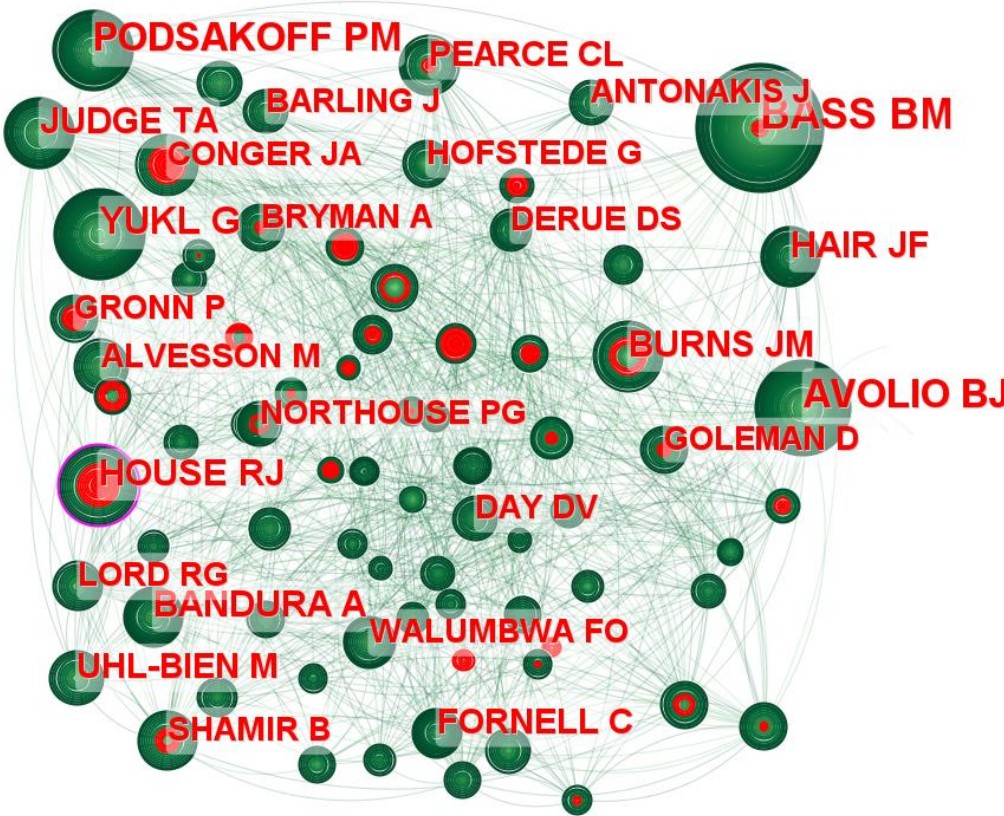

**Figure 7.** Author co-citation network.

**Table 4.** Top 10 authors in terms of co-citation frequency, burst strength and centrality.

| Top 10 Authors in Co-Citation Frequency | Top 10 Authors in Burst Strength | Top 10 Authors in Centrality |
| --- | --- | --- |
| Bernard M. Bass (541) | Jane M. Howell (14.62) | Robert J. House (0.17) |
| Bruce Avolio (363) | D.A. Aga (13.84) | Bernard M. Bass (0.1) |
| Philip M Podsakoff (308) | Jörg Henseler (13.79) | Alan Bryman (0.09) |
| Gary Yukl (283) | Fred E.Fiedler (12.51) | Bruce Avolio (0.07) |
| Joseph F. Hair (235) | Robert J. House (11.77) | James McGregor Burns (0.07) |
| Robert J. House (211) | Robert R. Blake (11.56) | Michael Fullan (0.07) |
| Timothy A. Judge (190) | Jay A. Conger (11.31) | Philip M Podsakoff (0.07) |
| James McGregor Burns (185) | Lianying Zhang (10.72) | Robert R. Blake (0.06) |
| Albert Bandura (178) | Daniel Katz (10.22) | Mats Alvesson (0.06) |
| Claes Fornell (162) | James R. Meindl (9.94) | Paul Hersey (0.06) |

According to the research topics of the co-citation authors, the distribution of their subject areas can be roughly divided into three categories: leadership theory (Bernard M. Bass, Bruce Avolio, Philip M Podsakoff, Gary Yukl, Robert J. House, Timothy A. Judge, James McGregor Burns, Jane M. Howell, James R.Meindl and Robert J. House), research methods (Joseph F. Hair, Claes Fornell, Jörg Henseler and Alan Bryman) and management (D.A. Aga, Robert R. Blake, Lianying Zhang and Paul Hersey). Keeping track of the research work of authors in the field of construction project management could facilitate capturing the latest developments in the field, while the studies of authors in the field of leadership theory and research methods could drive researchers to make theoretical and methodological innovations.

### 3.2.2. Reference Co-Citation Analysis

The reference co-citation network was generated by CiteSpace, using a log-likelihood ratio (LLR.) weighting algorithm to analyse the articles and their references in the data. A total of 102,546 distinct references were analysed in the process, and cluster labels were displayed once the process had been completed. The clusters were numbered in descending order of cluster size, starting from the largest cluster #0, the second-largest cluster #1 and so on. As shown in Figure 6, the intellectual structures of the construction knowledge domain were investigated. The brighter the colour of the area, the more active the field of research. The network of reference co-citations presented in Figure 8 contains 1163 nodes and 2943 links. According to Chen Chaomei, the developer of citespace, high-frequency co-cited articles are often regarded as milestones due to their creative contributions [30]. Therefore, the top 10 cited references listed in Table 5 form the theoretical basis for this knowledge domain.

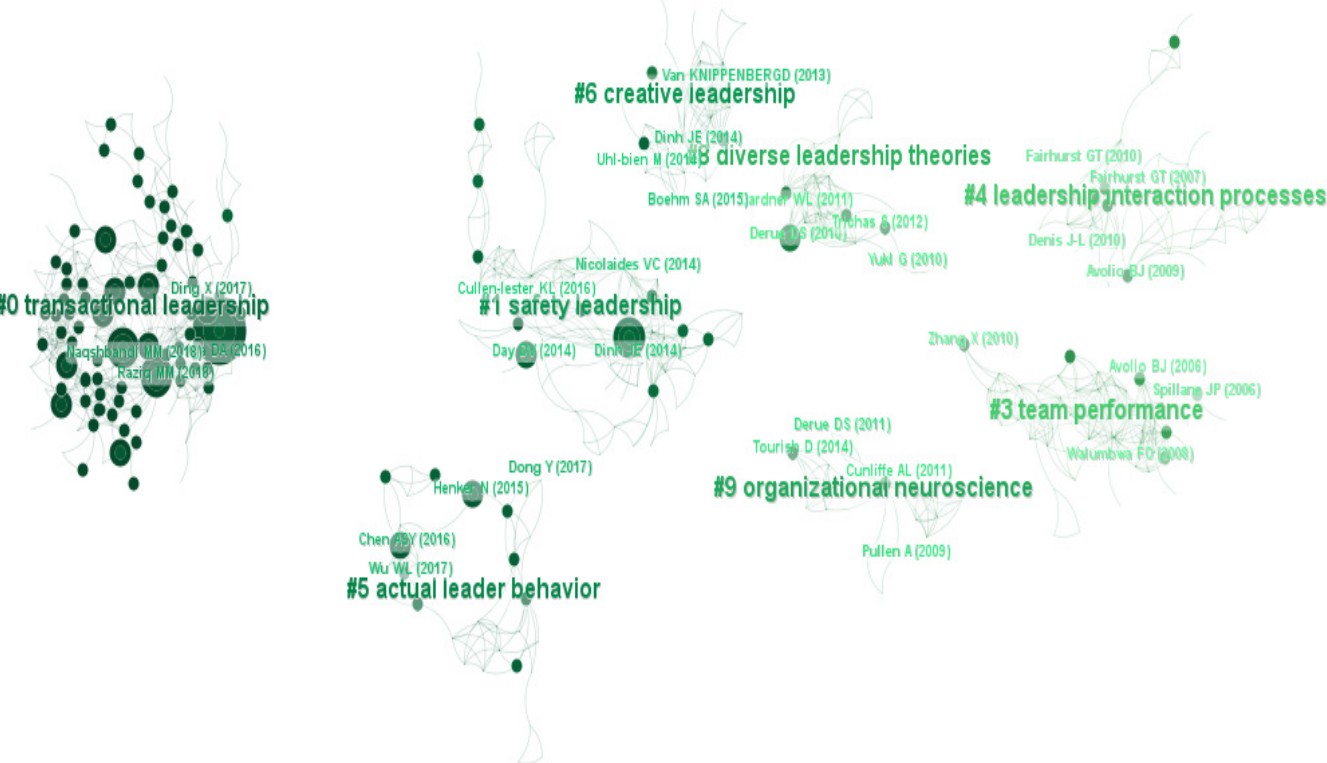

**Figure 8.** Reference co-citation network.

The network is divided into eight co-citation clusters based on the text analysis using the LLR. algorithm, which usually gives the best result in terms of uniqueness and coverage [44], and the information of the eight clusters is presented in Table 6. The silhouette number measures the average homogeneity of a cluster. All clusters received a high score, which indicated a good consistency of the cluster members. As shown in Table 5, the top terms of the clusters are traditional and long-lasting research topics in the construction project management field, which reflect the knowledge base of construction leadership research. In order to clarify the knowledge structure of this research field, a detailed discussion of the top five clusters is presented below. Clusters 6, 8 and 9 were also excluded due to their small size.

**Table 5.** Top 10 cited references.

| Citation Counts | References | Author | Source Journal/Publisher |
|---|---|---|---|
| 24 | Transformational leadership and project success: The mediating role of team-building [45] | D.A. Aga | International Journal of Project Management |
| 13 | Leadership theory and research in the new millennium: Current theoretical trends and changing perspectives [46] | Jessica E.Dinh | The Leadership Quarterly |
| 13 | Leadership styles, goal clarity, and project success: Evidence from project-based organisations in Pakistan [47] | Muhammad Mustafa Raziq | Leadership & Organization Development Journal |
| 11 | Knowledge-oriented leadership and open innovation: Role of knowledgemanagement capability in France-based multinationals [48] | M. Muzamil Naqshbandi | International Business Review |
| 10 | Linking transformational leadership and work outcomes in temporary organisations: A social identity approach [49] | Xian Ding | International Journal of Project Management |
| 10 | A primer on partial least squares structural equation modelling (PLS-SEM) [50] | Joseph F. Hair | Sage publications |
| 10 | Advances in leader and leadership development: A review of 25 years of research and theory [51] | Joseph F. Hair | Engineering, Construction and Architectural Management |
| 8 | Leadership, organisational culture, and innovative behavior in construction projects: The perspective of behavior-value congruence [52] | David V.Day | The Leadership Quarterly |
| 8 | Servant Leadership: A systematic review and call for future research [53] | Nthan Eva | The Leadership Quarterly |
| 8 | A meta-analytic review of authentic and transformational leadership: A test for redundancy [54] | George C.Banks | The Leadership Quarterly |

**Table 6.** Top eight co-citation clusters.

| Cluster ID | Size | Silhouette | Mean (Year) | Top Terms (Log-Likelihood Ratio) |
|---|---|---|---|---|
| 0 | 127 | 0.995 | 2021 | Transactional leadership |
| 1 | 77 | 0.927 | 2016 | Safety leadership |
| 3 | 52 | 0.938 | 2010 | Team performance |
| 4 | 37 | 0.981 | 2011 | Leadership interaction processes |
| 5 | 31 | 0.983 | 2018 | Actual leader behaviour |
| 6 | 29 | 0.954 | 2016 | Creative leadership |
| 8 | 22 | 0.981 | 2013 | Diverse leadership theories |
| 9 | 18 | 0.983 | 2014 | Organisational neuroscience |

(1) Transactional leadership

The "transactional leadership" cluster had the largest size in the construction leadership field, with a focus on the relationship between the leadership styles of project managers and project success. As a representative type, transactional leadership is used as a cluster label by the LLR algorithm. In the project management field, project success is a major concern and a recurring theme in the literature. Although there is still disagreement on what "project success" is, many leading scholars in the project management field agree with the point of view that the project manager is an important factor leading to project success [55]. Therefore, the factors related to project manager leadership have received the most attention in this field of research. In the two most co-cited papers in cluster 0, D.A. Aga [44] examined the mediating role of team building between transformational leadership and project success based on a field survey of 200 development project managers. At the same time, Muhammad Mustafa Raziq [46] contextualised transactional leadership

style and transformational leadership style in the project environment to clarify the impact of leadership style on project success.

Although cluster 0 has the largest size in the construction leadership field, there are still few studies directly exploring the relationship between leadership and program success. Zhang [56] explored the mediation role of leadership styles between project managers' emotional intelligence and other participants' satisfaction. Tabassi [57] proposed that the leadership competency of project managers is a critical factor in project success. Maqbool [58] conducted a quantitative study on the Pakistani construction industry, and the results show that project managers with desired leadership competencies ensure higher project success rates. Some other representative works of literature use leadership as a factor in research models [59–61]. An increased understanding of the factors that influence project success is very important for project-based organisations. Many factors affect the success of a project. The project manager is one of the most important of these factors. Therefore, project success as a criterion can provide a perspective on the leadership of project managers and can help in better understanding the mechanism of the project manager's leadership [44].

(2) Safety leadership

With technological improvements, significant advances have been made to safety levels in the construction industry. The focus on safety has shifted from individual work to organisations. Meanwhile, there has been a significant increase in research on leadership and safety [62]. For example, Hoffmeister [63] explored the links between individual leadership facets and the safety outcomes of the employee. Mullen [64] examined the moderating role of transformational leadership on the relationship between employer safety obligations and employee safety performance based on social exchange theory. Wu [65] identified four dimensions of safety leadership practices to match the types of leaders in construction projects.

Much of the research in this cluster focuses on the safety performance of individual workers [66–68]. However, the two most co-cited papers in this cluster developed multilevel models and introduced other research subjects. Wu [69] proposed that the safety leadership of owners, contractors and subcontractors in construction projects affected each other and clarified the impacting paths. Guo B H W [70] developed an integrative model to predict safety behaviour in the construction industry. Therefore, as the construction industry continues to develop, there is a need to develop a more comprehensive and integrated model exploring the mechanism of safety leadership in the construction industry.

(3) Team performance

Project teams are central to organisations. Due to the temporary nature of project teams, some project teams still cannot reach a stable and mature team when the project is completed, resulting in project failure [71]. Hence, the effective operation of the project team must be highly valued in construction projects [72].

The project team is composed of members with different demands, goals and expectations, and the team leader should motivate members in the early stage to form a team with consistent goals, vision and cohesion [73]. As a temporary organisation, project team members often have different professional backgrounds. Therefore, the project manager's ability to motivate members directly affects team performance [74]. Odusami et al. [8] ranked leadership and motivation as the most important skills for the project leader. Schmid & Adams [71] emphasised that team motivation can be heavily influenced by the project manager, especially during the early stages of the project. Ralf Müller & Turner [75]'s study came to a similar conclusion. Gehring [76] argued that the ability of project managers to perceive the motivations and demands of project team members is the basis of team leadership. Moreover, the competency of project managers in providing task guidance to project team members has been shown in empirical research to impact project performance significantly [75].

(4) Leadership interaction processes

Leadership is constructed and works in the process of practical interaction [77]. Meanwhile, leadership is assumed to positively contribute to most modern organisations' action processes [78]. In practice, different interaction styles of the leaders and followers always imply different group productivity levels [79]. Therefore, Carroll [25] urged leadership research on the "practice turn" in social settings by leaders and followers in interaction. In this cluster, scholars' interests focus on the leadership interaction between different project stakeholders. Chunlin Wu explored the leadership interaction between the key project stakeholders in his two studies [65,69]. Limao Zhang developed a model to clarify the relationships between different stakeholders' leadership and safety performance in a changing construction project environment. Furthermore, the interaction of the individual with the project environment is concerned in this cluster [51]. The study of leadership interaction is interdisciplinary [80], which requires collaborative research between scholars with different disciplinary backgrounds.

(5) Actual leader behaviour

The leader's behaviour can positively impact role clarity and work engagement [81]. Research in this cluster has focused on the impact of the project managers' behaviour on project team members. For example, Naoum [82] developed a model for the stress-coping behaviour of UK construction project managers. NannanWang [83] conducted comparative research to reveal how project managers' conflict-resolving behaviours affect project success. At the same time, Owusu-Manu [84] explored the linkages between the project manager's leadership style and mindset behaviour. The studies in this cluster offered a new, empirical insight into understanding the project manager's leadership.

### 3.3. Co-Word Analysis

### 3.3.1. Co-Occurrence Analysis

In order to trace research trends and topics, a timezone view of keywords co-occurrence analysis was conducted, and the result is visualised in Figure 9 below. The keyword co-occurrence network contained 558 nodes and 2679 links. The node size shows the frequency at which a keyword occurred in the data. The top 10 keywords in terms of co-occurrence frequency, burst strength and centrality are listed in Table 7. These keywords connected different research topics and significantly influenced the research of leadership in construction.

**Table 7.** Top 10 keywords in terms of co-occurrence frequency, burst strength and centrality.

| Top 10 Keywords in Co-Occurrence Frequency | Top 10 Keywords in Burst Strength | Top 10 Keywords in Centrality |
|---|---|---|
| Leadership (83) | Project management (7.38) | Construction (0.36) |
| Transformational leadership (73) | Mediating role (6.67) | Leadership (0.34) |
| Construction (66) | Leadership (5.37) | Construction industry (0.34) |
| Construction industry (65) | Model (4.62) | Behaviour (0.18) |
| Performance (63) | Construction project (4) | Construction project (0.13) |
| Behaviour (38) | Safety climate (4) | Transformational leadership (0.11) |
| Management (38) | Transactional leadership (3.95) | Management (0.1) |
| Project management (38) | Professional aspect (3.91) | Performance (0.09) |
| Model (30) | Construction industry (3.86) | Project management (0.08) |
| Mediating role (26) | Behaviour (3.8) | Model (0.05) |

As shown in Figure 10, the top 10 keywords with the strongest citation bursts were detected to explore the research field in more depth. The number of citations of the keyword "mediating role" has increased dramatically since 2020, with a high strength number of 6.67, indicating the direction of this research field in the next few years. Academics are becoming increasingly concerned with the mediating role between leadership and project success.

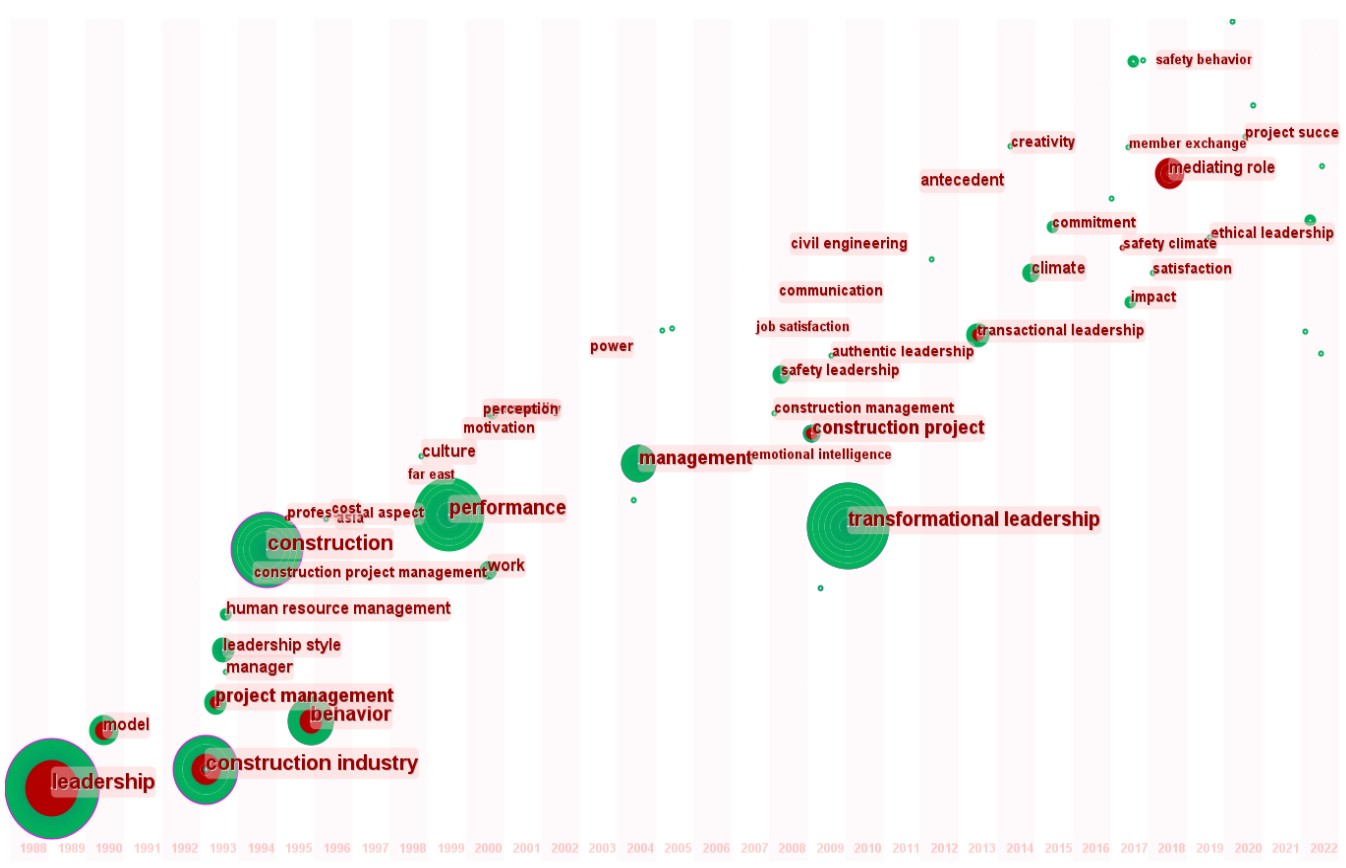

**Figure 9.** Keywords co-occurrence network from a timezone view.

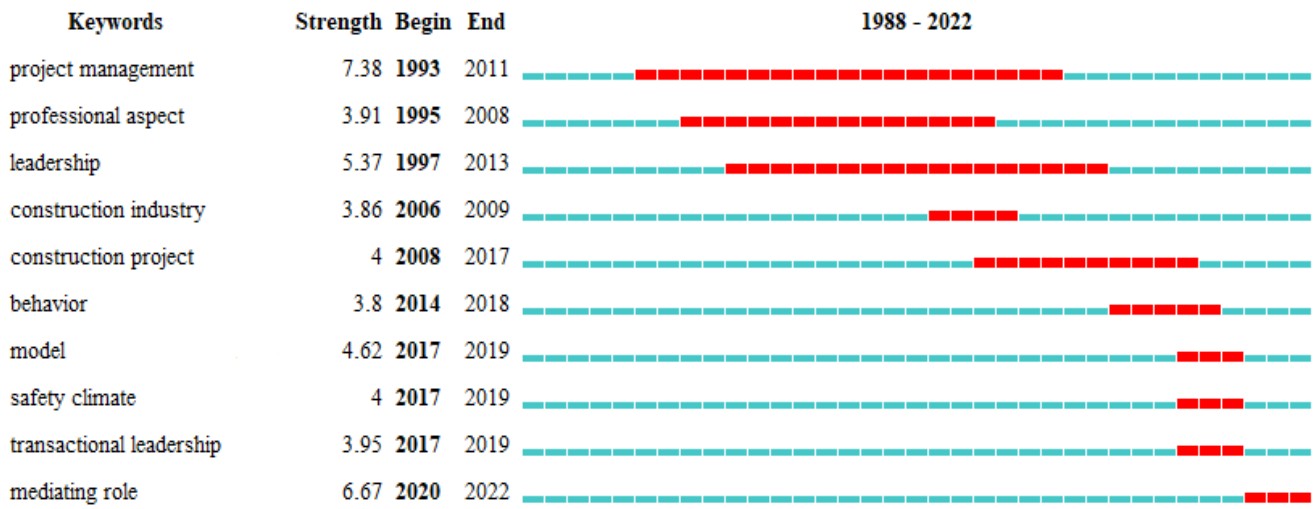

**Figure 10.** Top 10 keywords with the strongest citation bursts.

### 3.3.2. Keywords Evolution Analysis

The keywords evolution trend was also shown in the timezone view of co-occurrence analysis. In 1989, the keywords "leadership" and "model" appeared and marked the beginning of this field of study. During 1993–2000, there was a big outbreak of keywords, including "construction", "construction project management", "performance", "leadership style", "behaviour", "culture" and "perception", with a high co-occurrence frequency.

During 2008–2013, the keywords "transformational leadership" and "transactional leadership" appeared and gradually became a research hotspot. There are new co-occurrence keywords, such as "stakeholder management" and "knowledge sharing", which appeared with no highly frequent one in the last two years due to the short time for accumulation. Furthermore, the appearance of the keyword "PLS-SEM" in 2022 is noteworthy. As a modern multivariate analysis technique, the application of PLS-SEM is relatively new in the construction management field [85].

## 4. Future Research Prospects

With a large and rapidly growing body of leadership research, the development of related research in construction has led to many achievements. Based on the research results of Section 3, we propose the following trends and innovative research areas for future research.

### 4.1. Future Reserch Trends

(1) More studies on leadership in different countries. According to the collaboration analysis, it can be seen that new countries and regions enter this research field every year. Due to the variable level of the progress of urbanisation in various countries, the construction industry in developing countries is gradually attracting research attention. Researchers should expand the geographic scope of the countries studied to discover differences in the construction industry between less developed and developed countries. In the process, cross-country academic cooperation should be gradually strengthened. The impact of policy and cultural differences across countries on leadership in the construction industry is worth examining. In practice, these explorations are of great significance for international cooperative construction projects.

(2) More different theoretical perspectives. Leadership theory has practice-focused origins and is designed to address practical challenges for leaders within an organisation [86]. With the development of leadership theory and organisational theory, we argue that researchers should consider the leadership of construction from different theoretical perspectives. We also need to think more broadly about the factors contributing to the improvement of leadership in construction. Therefore, stakeholders outside the project team will gradually be paid attention to by researchers. The emergence of the co-occurrence word, stakeholder management, in 2021 confirms this.

(3) More emphasis on the leadership behaviours of project managers. The growing use of the term "behaviour" shows that scholars are becoming increasingly concerned with how actual leadership behaviour affects the success or failure of a project. In the top eight co-citation clusters, cluster 5 focuses on the impact of the project managers' behaviour on project team performance. Research on leadership competencies in construction projects largely mirrors the evolution of the broader discourse of behavioural psychology and has attracted scholars' attention in recent years. It will be valuable for future research to identify the key leadership behaviours of a project manager that lead to project success. In practice, project managers can be quickly and accurately identified based on this criterion.

(4) More quantitative studies using the accurate and effective model. Leadership theory has practice-focused origins. The burst of the keyword "model" in 2017–2019 and the appearance of the keyword "PLS-SEM" in 2022 indicates the continuous emergence of quantitative research in the construction leadership field. Therefore, with in-depth study, the burst of the keyword "mediating role" is logical. Future research should integrate with existing research clusters, such as safety climate [87–89] and knowledge management [90], to conduct more objective empirical research on the mechanism of construction project organisation. The mediating role of project success factors when it comes to different types of leadership will become a research hotspot in the future [91–93].

*4.2. Innovative Future Research Areas*

Project-based work is booming around the world today. The world itself is constantly changing, and climate change, demographic shifts or pandemic diseases will have a profound impact on the organisation of human society. Therefore, the study of leadership in the project organisation is bound to generate new areas of knowledge with these changes.

(1) Demographic shift and leadership. Research on the relationship between leadership and demographic factors is not uncommon. Previous research has investigated the relationship between demographic traits and leadership from various perspectives, such as gender, nationality and educational background [94]. However, as the demographics have changed, not only have the characteristics of the project manager changed, but the project members are no longer the same. As baby boomers gradually retire, the demographic structure of the project organisation has changed, with practitioners showing the characteristics of youth, a strong personality, high education and a strong desire to learn. How to better manage these new generations of knowledge-based employees is a problem that must be solved in project management today. At the same time, gender issues are increasingly relevant in project management as more women enter the industry. Therefore, the role of leadership in project organisations with changing demographics can be an interesting area of research.

(2) Technology advances and hybrid leadership. As many new technologies, such as Building Information Modeling, big data or virtual teams, are applied to project management, the form of project organisation inevitably changes. Meanwhile, remote work has become the norm in the years since the COVID-19 pandemic. Therefore, as technology advances, a hybrid work environment will become a problem for every project manager. Hybrid environments are hybrid, highly flexible environments that combine traditional face-to-face environments with remote work environments, where employees move in and out of the office flexibly and rely heavily on technology-mediated communications. How do technology and virtual teams affect leadership styles, communication styles and project performance? The answers to this question will become an innovative area of research.

(3) Ethical leadership. Research on ethical leadership and corporate social responsibility has a long history. However, research on project ethics is still in its infancy. Ethical leadership contributes to the effective functioning of an organisation and also affects the attitudes, behaviours and performance of employees [95]. Research on leadership in project-based organisations can not only improve project performance but also drive the development of relevant industry standards.

## 5. Conclusions

This review sought to explore the current state and trends in leadership research in the field of construction. A large number of publications were considered, from the beginning of the study of construction leadership to the present. A scientometric analysis, based on CiteSpace, of a total of 1789 bibliographic records collected from Scopus and the WoS core collection database was conducted to identify and visualise the intellectual structures of the leadership knowledge domain in the field of construction. In construction leadership research, very few studies have addressed the theoretical changes over time. This review solved this problem by conducting a comprehensive analysis of the development of leadership theory in construction. Meanwhile, further systematic and accurate predictions of changes are provided in this field of study, which could guide scholars in this field and gain insights into future development.

The conclusions that can be drawn from the results are as follows. Since 2006, research on the leadership of the construction industry has gradually deepened, and the number of annual publications has increased significantly. The United States had the earliest start in this field, with a publication frequency and centrality obviously ahead of other countries. The US has also played an essential role in establishing international cooperation for related research. Research in related fields in China and Australia is also developing rapidly, but their performance in terms of international cooperation, which needs to be established gradually over a long time, needs to be strengthened. Scholars from new countries join this

research field every year, which has manifested in uneven global economic development. Most of the most prolific institutions and authors come from Asia and Australia, but the cooperation network between them is not well established. Scholars usually focus on the construction industry in their own country. The lack of international cooperation can also be seen from the co-citation analysis, which shows that cross-cultural and international project research has not become a significant cluster. Therefore, we propose that, in future research, scholars should pay more attention to the construction industries in developing countries and carry out more international academic cooperation.

According to the co-occurrence analysis of keywords and the article cluster analysis, it can be seen that transactional leadership, safety leadership, team performance, leadership interaction processes and actual leader behaviour formed related research clusters. Safety and performance are always of concern to researchers. Among the various leadership styles, transformational leadership has received the most attention. The keyword "mediating role" has suddenly appeared in a large number of articles in the past two years. It can be predicted that there will be more empirical studies on the relationship between leadership and project success in future research. At the same time, the mediating role of project success factors on different types of leadership should receive more attention from scholars. Therefore, we proposed following the potential trends: (1) More studies on leadership in different countries; (2) More different theoretical perspectives; (3) More emphasis on the leadership behaviours of project managers; (4) More quantitative studies using the accurate and effective model. At the same time, we also propose three innovative future research areas: (1) Demographic shift and leadership; (2) Technology advances and hybrid leadership; (3) Ethical leadership.

The contribution of this study is in the following two aspects. Theoretically, this article proposes a knowledge map to comprehensively depict the past, present and future of construction leadership research. The collaboration analysis reveals the main contributors in this domain at a micro level. The co-citation analysis identifies the critical research areas, reflecting the knowledge bases and research hotspots. Moreover, the co-word analysis traces the knowledge evolution and presents the historical research hotspots. The results of this study not only point to future research trends but also provide a reference for construction companies to formulate project manager selection criteria at the practical level. Limited by CiteSpace and language, the samples in this study are peer-reviewed articles in English from Scopus and the WoS core collection database. Increasing the range of data sources in future research may result in a more accurate intellectual structure of the leadership knowledge domain in the field of construction. Moreover, the use of other visualisation techniques should be encouraged to enrich the map of this knowledge domain.

**Author Contributions:** Conceptualisation, W.P.; methodology, W.P.; software, W.P.; writing—original draft preparation, W.P.; writing—review and editing, N.A.H.; supervision and review of the article, N.A.H.; advice and support, A.H.A. and T.H.L. All authors have read and agreed to the published version of the manuscript.

**Funding:** This research received no external funding.

**Data Availability Statement:** Not applicable.

**Acknowledgments:** The authors would like to thank the Department of Civil Engineering, Faculty of Engineering, University Putra Malaysia, Serdang, Malaysia, for supporting this review paper.

**Conflicts of Interest:** The authors declare no conflict of interest.

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
