# Peer review of "Leadership in Construction: A Scientometric Review"

_buildings, doi:10.3390/buildings12111825_

Round 1
Reviewer 1 Report (Previous Reviewer 2)
Some suggestions to improve the quality of the article:
- The authors should explain the difference between competence, skill, and ability. For example, row 55: personality is not a competence.
- Why did the authors exclude 444 articles?
- The authors should explain why they selected the articles from 1990 to 2022. Is there an evolution on project managers` competences? Did they find changes in the construction area? E.g. Was innovation important in 1990?
- Is there a connection between leadership ant the project`s size? Please explain.
Author Response
Please see the attachment.

Reviewer 2 Report (New Reviewer)
Do not think this manuscript has any innovation and contribution. it is rejected on my side without any chance to modify.
Author Response
Please see the attachment.

Reviewer 3 Report (New Reviewer)
The authors present an interesting study focusing on leadership in construction using a scientometric review analysis approach. While the reviewer thanks the authors for reporting the current state of work in this field and the allied future directions, there are still significant changes required in the manuscript to impart a certain rigor required for a journal publication.
The reviewer's primary concern is the primary contributions of the work. There is no paucity of literature in the area of leadership and its related studies in the realm of construction industry. How is the current manuscript filling a gap by using a scientometric analysis? And how is this method (scientometric) better than the previous review studies?
Another important deficiency the reviewer notes is the use of just one source of data - Web of Science. Without a doubt, WoS is a powerful database, however, Scopus is also equally powerful as it pertains to leadership and thus should be included in the current work so as to not miss out on other important literature. There will be duplicates and need to be manually cleaned to yield enhanced sources from the literature.
Another common problem noticed is the references are not numbered uniformly where there are multiple references present. For example, some references are numbered [n][m], some others are presented as [n-m]. This may be rectified throughout the manuscript.
The authors are highly encouraged to perform a thorough proofreading of the next version of the manuscript to rectify minor spelling mistakes present in the current manuscript.
Apart from the general comments, here are a few specific section-wise comments that can help improve the manuscript.
Abstract
Ln 18 – Scholors should be scholars.
Reason to perform the study is unclear – is it just because there is a paucity of literature in the area of leadership as it pertains to construction (Ln 9)? Or is (are) there any other motivating factor(s)? Scientometrics is just a tool being used, but what the actual contributions of the manuscript are needs to be made clear in this section and other sections as well.
Introduction
Ln 61 – what are these ‘other aspects’?
Ln 73 – 75 – Can the authors justify that ‘quantitative bibliometric analysis in this field’ is better than subjective expert opinions? If so, how?
Ln 92 – The authors must make it clear that the lack of research relates to scientometric analysis of construction leadership.
Results
Ln 212 – Remove the column betweenness centrality and add a note that all authors have a zero value for this measure in the title of the table.
Ln 215 – notes should be nodes
Ln 254-255 – Reference 31 refers to a biological review paper and I am assuming the authors of the current manuscript are trying to imply the relative importance of the top 10 works as highlighted by reference 31. If this is the case, the same must be made amply clear to the reader.
Table 4 – List all the authors of each paper presented in this table in a separate column.
Table 5 – If Green Buildings does not relate to this study (Ln 269), why is it listed here as Cluster 2? Moreover, why have the results not been removed from the number of relevant papers?
Green Buildings is referred to as Cluster 3 in Ln 267 whereas in Table 5, it is listed as Cluster 2. This requires appropriate change as needed.
Ln 352 – notes
Ln 353-360 – Can these not be represented in a tabular format?
Figure 9 – What does the column ‘Year’ with a common value 1992 signify? Does this warrant a removal?
The reviewer wishes the authors the best for the next round of the manuscript.
Round 2
Reviewer 3 Report (New Reviewer)
The authors seem to have significantly improved the quality of the paper and I feel it is ready for publication.
Author Response
Please see the attachment.

This manuscript is a resubmission of an earlier submission. The following is a list of the peer review reports and author responses from that submission.
Round 1
Reviewer 1 Report
This manuscript adopted Scientometric method and Citespace as a tool to conduct a review of 1049 bibliographic record retrieved from WOS related to “leadership in Construction” topic. Reviewer found the manuscript interesting. This manuscript perhaps can be improved from the following aspects:
Major
- can you add a section to summarize the future trends of “leadership in construction”
- significance and academic and practical contributions of the study should be discussed in the discussion section and highlighted in the conclusions section and reflect in abstract.
- Include a diagram to explain the overall research method, you may refer to some of publications:
- https://doi.org/10.1016/j.jclepro.2020.125223
- https://doi.org/10.1016/j.ijproman.2016.08.001
- https://doi.org/10.3390/buildings9040085
- https://doi.org/10.3390/buildings9100210
- can you clarify the difference between your review and other relevant studies? How can your study bridge the gaps in the relevant review studies.
- The author has reviewed global leadership in Construction, should proposed future research directions and current research hotspots in conclusion.
- Extensive editing of English language and style required.
Minor
- Abstract - Abstract section is too short, authors should consider including the contribution after scientometric analysis, especially what new knowledge is generated from this study?
- Abstract – What is the research motivation and research gaps? Why this study is important?
- Keywords – Rearrange keyword alphabetically
- Table 3 – no author with centrality? Can you explain?
- Figure 2 and 3 – should be country/region. Authors pls check other figures.
- Figure 4 – missing colour legend.
Reviewer 2 Report
Some suggestions to improve the quality of the article:
- The authors refer to qualitative analyze in previous papers regarding the leadership style. What do they mean? There are quantitative results as well. E.g. Bakar, A. (2010). Towards Assessing the Leadership Style and Quality of Transformational Leadership: The Case of Construction Firms of Iran, Journal of Technology Management, China; Liphadzia, M., Aigbavboab, C., Thwalac, W. (2015). Relationship Between Leadership Styles and Project Success in the South Africa Construction Industry, Elsevier, Procedia Engineering 123.
- What is the contribution of the authors, other than taking information from Internet? Is a new statistic a scientific result?
- The authors refer to more than 1000 articles, but the reference section is very poor.
Round 2
Reviewer 1 Report
- The authors do not seem to have addressed all the first round of comments. I urge that authors review first round of comments.
- I am not convinced with the future research direction.
- The entire manuscript looks like a report made by citespace tool, lacks scientific soundness and seems no scientific contribution.
Reviewer 2 Report
The authors didn`t make any changes. They just explained what bibliometric research is, which, of course, I know.
They say that only quantitative results exist in the literature, which is not correct. I gave them examples of qualitative analyze.
I asked them which is the difference between their statistics and others. No strong answer.
I told them that reference is poor, although they mentioned more than 1000 articles in the study. They answered that they used them for statistics. Not even read some of them?